# Angular Super-Resolution in Diffusion MRI with a 3D Recurrent Convolutional Autoencoder

**Matthew Lyon**[1]                                              MATTHEW.LYON-3@POSTGRAD.MANCHESTER.AC.UK
**Paul Armitage**[2]                                                              P.ARMITAGE@SHEFFIELD.AC.UK
**Mauricio A Álvarez**[1]                                      MAURICIO.ALVAREZLOPEZ@MANCHESTER.AC.UK

[1] *Dept. of Computer Science, University of Manchester, UK*

[2] *Dept. of Infection, Immunity and Cardiovascular Disease, University of Sheffield, UK*

## Abstract

High resolution diffusion MRI (dMRI) data is often constrained by limited scanning time in clinical settings, thus restricting the use of downstream analysis techniques that would otherwise be available. In this work we develop a 3D recurrent convolutional neural network (RCNN) capable of super-resolving dMRI volumes in the angular (q-space) domain. Our approach formulates the task of angular super-resolution as a patch-wise regression using a 3D autoencoder conditioned on target b-vectors. Within the network we use a convolutional long short term memory (ConvLSTM) cell to model the relationship between q-space samples. We compare model performance against a baseline spherical harmonic interpolation and a 1D variant of the model architecture. We show that the 3D model has the lowest error rates across different subsampling schemes and b-values. The relative performance of the 3D RCNN is greatest in the very low angular resolution domain. Code for this project is available at github.com/m-lyon/dMRI-RCNN.

**Keywords:** Diffusion MRI, Deep Learning, Angular super-resolution, Recurrent CNN, Image Synthesis

## 1. Introduction

Advances in diffusion MRI (dMRI) analysis techniques continue to push the boundaries of what is attainable through the non-invasive imaging modality (Zhang et al., 2012; Raffelt et al., 2017; Drake-Pérez et al., 2018). However, acquiring the high angular resolution diffusion imaging (HARDI) that is needed for these more advanced techniques presents a challenge. HARDI data requires the acquisition of typically thirty or more diffusion directions, often at several b-values (multi-shell), to use these techniques effectively. It is therefore clinically infeasible to benefit from these advances due to the time constraints of acquiring such high resolution datasets.

One way to reduce the burden of acquisition time is through the use of image enhancement techniques such as super-resolution (SR). Here dMRI data has two distinct, but related, resolutions that can be super resolved: spatial resolution, or the density of sampling within k-space, and angular resolution, or the density of sampling within q-space (Tuch, 2004). Spatial super-resolution (SSR) has been extensively covered in the medical imaging and natural image domains (Li et al., 2021; Yang et al., 2019). However, as dMRI data has a unique angular structure, the amount of work done in the angular super-resolution (ASR) domain is relatively limited.

In particular, many methodologies opt to constrain the challenges of ASR by performing inference on downstream analysis techniques. This has the advantage of simplifying the task, but limits the ability of the super-resolved data to be used in different analysis techniques. Both Lucena et al. (2020) and Zeng et al. (2021) used single-shell data and convolutional neural network (CNN) architectures to infer fibre orientation distribution (FOD) data with similar quality to a multi-shell acquisition (Tournier et al., 2007). Similarly, Golkov et al. (2016), Chen et al. (2020), and Ye et al. (2020) developed deep architectures to infer metrics from models such as neurite orientation dispersion and density imaging (NODDI) (Zhang et al., 2012) and others, that would otherwise be unavailable with single-shell data.

Models that work with diffusion data directly often do so with the use of spherical harmonics (SH) (Frank, 2002). SH provide a set of smooth basis functions defined on the surface of a sphere. As they form a complete orthonormal basis, they can be used to describe any well behaving spherical function. As such, they are commonly used to represent dMRI signal, which is measured at different diffusion directions defined by points (b-vectors) on the surface of a unit sphere. Typically within dMRI deep learning the SH coefficients are first fit to the diffusion data, and then used as input in place of the unconstrained diffusion signal. The network can then be trained to infer the SH coefficients of other shells, as the SH framework already provides interpolation to other data points within a single shell. For example, Koppers et al. (2016) used SH coefficients from single-shell dMRI data to infer SH coefficients of a different shell. This method was limited in scope however, as only randomly sampled white matter (WM) voxels within the brain were used. Jha et al. (2020) then extended this idea using a 2D CNN autoencoder architecture, that inferred data across the whole brain. Currently only one other deep learning architecture proposed by Yin et al. (2019) infers raw dMRI data without the use of SH. This architecture is a 1D CNN autoencoder, which therefore does not benefit from the spatial relationships present within dMRI data.

This paper proposes a novel implementation of ASR in dMRI data through the use of a recurrent CNN (RCNN) autoencoder architecture. This involves two key innovations: 1) extending the dimensionality of the network to 3D; 2) using a 3D convolutional long short term memory (ConvLSTM) cell to model the q-space relationships. Both of these contributions allow us to leverage the spatial correlations present within dMRI data to efficiently infer new diffusion directions without the constraint of predefined functions such as SH. Additionally, omitting the SH framework allows us to explore the feasibility of using unconstrained dMRI data in deep learning inference.

We evaluate the performance of the proposed model by measuring the deviation of dMRI signal from the ground truth across multiple diffusion directions. The WU-Minn Human Connectome Project (HCP) dataset (Van Essen et al., 2013) is used for training and quantitative comparison. We evaluate model performance across different angular resolutions and b-values. Additionally, we compare results from our proposed 3D model with angular interpolation within the SH framework, and a 1D variant of the same model.

## 2. Methods

We formulate the task of ASR in the following way: low angular resolution (LAR) datasets, comprising of 3D dMRI volumes and b-vectors, are used as context data to generate a latent

representation of the entire q-space. This latent representation is then queried with target b-vectors, to infer previously unseen dMRI volumes. We list below the pre-processing steps required and network implementation.

## 2.1. Pre-processing

The HCP dMRI data is used for both training and evaluation, and is initially processed with the standard HCP pre-processing pipeline (Glasser et al., 2013). Each 4D dMRI volume within each subject in the HCP dataset contains three shells of b-values 1000, 2000, and 3000. Each shell is processed independently and contains 90 diffusion directions, of which the LAR dataset are subsampled from. Several further pre-processing steps are necessary to transform the data into an appropriate format for efficient training within the network. First dMRI data are denoised. Noise is assumed to be independent across the entire 4D volume, therefore noise within the context dataset cannot be used to predict the noise within the target volumes. To mitigate this problem, a full-rank locally linear denoising algorithm known as 'patch2self' (Fadnavis et al., 2020) is applied to the data. Next, dMRI data are rescaled such that the majority of the distribution lies between $[0, 1]$. This is done by dividing each voxel by a normalisation value given it's shell membership. 4000, 3000, and 2000 were found to work well as normalisation values for the shells $b = 1000$, $b = 2000$, and $b = 3000$ respectively.

Afterwards, dMRI data are split into smaller patches with spatial dimensions $(10 \times 10 \times 10)$. This is done to mitigate the memory limitations of using 4D data, which would have a prohibitively large memory requirement if kept at full size. $10^3$ isotropic was found to be a large enough patch size to benefit from non pointwise convolutions, whilst still having reasonable memory requirements. Each patch contained at least one voxel from a brain extracted mask, thus patches which contained no voxels within the brain are discarded.

Next, during training only, the q-space dimension within dMRI patches and b-vectors are shuffled. This is a crucial step in encouraging the model to learn the relationship between the measured dMRI signal and the b-vector directionality, whilst additionally discouraging the model from converging on a solution that is sensitive to the order of the q-space samples. As such, the shuffling process is repeated after each training epoch. As the training examples are subsampled from the full 90 directions, it is important to ensure that the q-space shuffling produces examples that are approximately evenly distributed across the q-space sphere. To do this, an initial direction is chosen at random, then, the next points are sequentially chosen to minimise the total angular distance between all points previously selected. This is repeated until the number of points equals the training example size. Finally, the shuffled datasets are split into context and target sets of size $q_{in}$ and $q_{out}$ respectively.

## 2.2. Proposed Network

The proposed RCNN is a conditional autoencoder, comprising of a 3D CNN encoder and decoder. A graphical representation of the architecture is shown in Figure 1.

### Encoder

Each encoder input consists of a context set, of size $q_{in}$, containing dMRI patches and corresponding b-vectors. Initially within the encoder, the set of b-vectors are repeated in

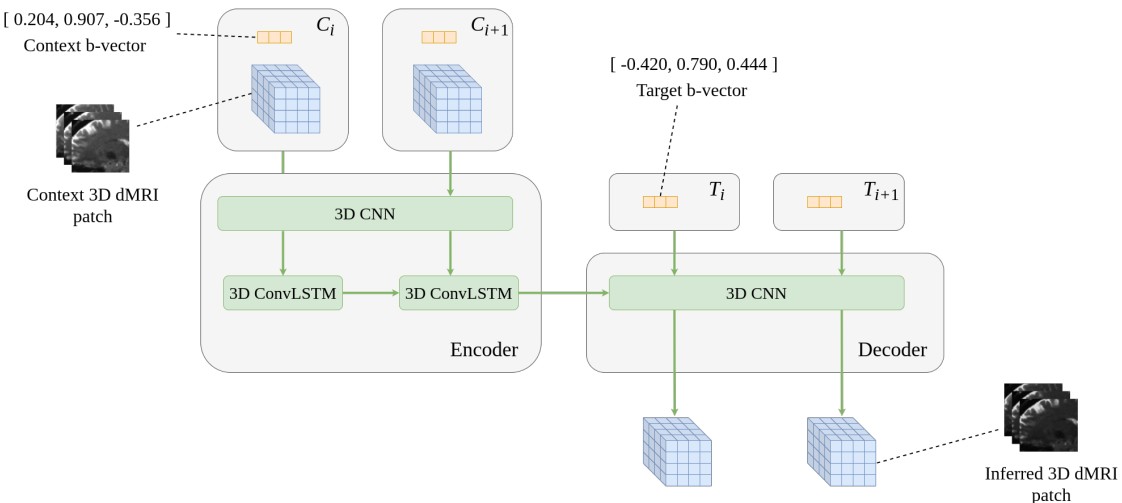

Figure 1: RCNN model design. Here q-space context data $C_i$ are given to the encoder sequentially until all context examples $C_{q_{in}}$ are seen. Next, the internal hidden state of the ConvLSTM is passed to the 3D CNN decoder along with target data $T_i$ to infer 3D dMRI patches along the given diffusion direction.

each spatial dimension, then concatenated with the dMRI signal, channel-wise, to form a 'q-space tensor'. Next, the q-space tensor is passed through a pointwise convolutional layer, then onto two parallel convolution blocks connected in series. Afterwards, the output is passed through two more pointwise convolutional layers, and finally onto a 3D ConvLSTM layer. As the convolutional layers within the encoder contain both dMRI signal and b-vector information, this allows the encoder network to directly learn the q-space representation of the dMRI data.

## DECODER

The hidden internal state within the ConvLSTM layer, alongside a set of target b-vectors, is used as input to the decoder. First, the hidden state is repeated $q_{out}$ times, where $q_{out}$ is the number of target b-vectors. Then, similarly to the encoder, the target b-vectors are repeated in each spatial dimension and concatenated with the hidden state channel-wise. Afterwards the resultant tensor is passed through two pointwise convolutional layers, then subsequently concatenated with the target b-vectors again. Finally this is passed through two parallel convolutional blocks, and subsequently two convolutional layers.

## PARALLEL CONVOLUTION BLOCKS

Parallel convolution blocks are used within the encoder and decoder. They consist of three convolution layers with kernel sizes $(1 \times 1 \times 1)$, $(2 \times 2 \times 2)$, and $(3 \times 3 \times 3)$, that apply a convolution operation to the same input in parallel. The $2^3$ and $3^3$ isotropic kernels have padding applied prior to the convolution operation, such that the resultant shape is equal to

the unpadded input tensor. This block is inspired by work done in Szegedy et al. (2016), and allows for different resolutions of spatial information to pass through the block in tandem. Outputs from the three convolutions are then concatenated together channel-wise with an additional residual input. Within the encoder, this residual input is the aforementioned q-space tensor, and within the decoder it is the target b-vectors that have been repeated across spatial dimensions. By including this additional input, b-vector directionality and image information has straightforward propagation throughout the network, allowing the model to more easily learn the complex q-space relationship within the data.

IMPLEMENTATION DETAILS

All convolutional layers, excluding the ConvLSTM layer and the final two convolutional layers within the decoder, consist of the following: a convolution operation, a Swish activation function (Ramachandran et al., 2017), and either instance or batch normalisation. The final two decoder layers have no normalisation and, as dMRI is strictly positive, Rectified Linear Unit (ReLU) activation is used in the final layer in place of a Swish activation. The ConvLSTM is a 3D extension of the 2D ConvLSTM presented in Shi et al. (2015), where the dense connections of a standard long short term memory (LSTM) cell are replaced with convolutional kernels. It uses standard activations found within a LSTM cell and no normalisation. Each of the convolutional layers, except for the ConvLSTM layer, treat the q-space dimension as an additional batch dimension, therefore sharing parameters across q-space samples. All convolutional layers use a stride of $(1 \times 1 \times 1)$. Hyperparameters used for each layer can be found in Figure A.1, and were obtained by a hyperparameter search using KerasTuner (O'Malley et al., 2019) and the Hyperband algorithm (Li et al., 2017). Models were trained for 120 epochs using the optimizer Adam (Kingma and Ba, 2014) with mean absolute error (MAE) loss function and a learning rate of 0.001. The weights used for analysis were the best performing within the validation dataset. Training and validation datasets comprised of data from 27 and 3 HCP subjects, respectively.

### 2.3. SH Q-Space Interpolation

To interpolate dMRI data using SH, first SH coefficients $\mathbf{c}_{sh}$ for each spatial voxel are found using the pseudo-inverse least squares method in equation (1) below,

$$\mathbf{c}_{sh} = (\mathbf{B}_L^\top \mathbf{B}_L)^{-1} \mathbf{B}_L^\top \mathbf{s}_L. \tag{1}$$

Here, $\mathbf{B}_L$ is a matrix that denotes the SH basis for the low angular resolution dataset, where each row contains the SH expansion sampled at a given diffusion direction. $\mathbf{s}_L$ is a vector containing the measured signal voxel at various diffusion directions. The full resolution dataset $\mathbf{s}_H$ is reconstructed simply via equation (2), where $\mathbf{B}_H$ is the SH basis matrix containing all diffusion directions within the shell,

$$\mathbf{s}_H = \mathbf{B}_H \mathbf{c}_{sh}. \tag{2}$$

The set of SH basis functions sampled to obtain $\mathbf{B}_L$ and $\mathbf{B}_H$ are the modified SH functions $\widetilde{Y}_l^m(\theta, \phi)$ first defined in Tournier et al. (2007):

$$\widetilde{Y}_l^m(\theta, \phi) = \begin{cases} 0 & \text{if } l \text{ is odd,} \\ \sqrt{2}\,\Im(Y_l^{-m}(\theta, \phi)) & \text{if } m < 0, \\ Y_l^0(\theta, \phi) & \text{if } m = 0, \\ \sqrt{2}\,\Re(Y_l^m(\theta, \phi)) & \text{if } m > 0, \end{cases} \tag{3}$$

where $Y_l^m(\theta, \phi)$ defines the SH basis function of order $l$ and $m$.

## 3. Experiments and Results

We evaluate the performance of our model on eight previously unseen subjects from the HCP dataset for three diffusion shells and at varying q-space undersampling ratios. Each result in the presented tables is obtained from a separately trained model, with the same architecture and hyperparameters, except for the 3D RCNN (Combined) model. Instead, this model is trained with all three b-values concurrently. As a baseline comparison, results from the RCNN models are compared to SH interpolation. Here, a maximum SH order of 2 is used, as this was found to produce the most accurate reconstruction within the subsampling ratios used. Root mean squared error (RMSE), and mean structural similarity index measure (MSSIM) results are presented with respect to the ground truth of the measured diffusion directions for the eight subjects. RMSE and MSSIM in the presented tables are given as a mean and standard deviation across q-space samples in the eight subjects. Each value within these error distributions is averaged across all spatial dimensions within each q-space sample and subject.

Table 1 compares performance at different subsampling ratios across the three ASR models. The proposed 3D RCNN outperforms both the 1D variant and SH interpolation across all three subsampling ratios in RMSE and MSSIM. The largest relative gain in performance is present at the lowest subsampling ratio $q_{in} = 6$. Here the 3D RCNN has a reduction in RMSE of 34.1% compared to SH interpolation, and a reduction in the RMSE standard deviation of 72.4%. This suggests that the 3D RCNN model is able to effectively leverage the relationships between neighbouring voxels within a patch. Notably, the relative performance of the 3D RCNN decreases with increasing subsampling ratios, compared to SH interpolation. This indicates that within a low subsampling regime, the learned joint kq-space distribution affords the 3D model additional information, not present in the q-space distribution. In a higher sampling regime however, the additional information present within the joint distribution is relatively diminished. This implies that past a certain threshold the q-space distribution alone can be used to effectively interpolate between points, without the need of additional spatial information provided by the 3D model.

| Model | $q_{in} = 6$, $q_{out} = 84$ | | $q_{in} = 10$, $q_{out} = 80$ | | $q_{in} = 30$, $q_{out} = 60$ | |
| --- | --- | --- | --- | --- | --- | --- |
| | RMSE | MSSIM | RMSE | MSSIM | RMSE | MSSIM |
| SH Interpolation | $119.0 \pm 50.3$ | $0.9460 \pm 0.0419$ | $65.1 \pm 12.7$ | $0.9854 \pm 0.0044$ | $63.8 \pm 10.4$ | $0.9867 \pm 0.0033$ |
| 1D RCNN | $102.5 \pm 31.6$ | $0.9639 \pm 0.0225$ | $70.0 \pm 15.1$ | $0.9852 \pm 0.0054$ | $64.1 \pm 10.6$ | $0.9875 \pm 0.0033$ |
| 3D RCNN | $\mathbf{78.4 \pm 13.9}$ | $\mathbf{0.9787 \pm 0.0071}$ | $\mathbf{63.4 \pm 12.5}$ | $\mathbf{0.9870 \pm 0.0040}$ | $\mathbf{63.4 \pm 10.2}$ | $\mathbf{0.9876 \pm 0.0032}$ |

Table 1: Average performance of ASR in eight subjects with $b = 1000$ across different models. Best results are highlighted in bold.

Table 2 similarly shows that optimal performance is obtained from the individually trained 3D RCNN models, this time across all b-value shells. Additionally the combined 3D model outperforms the 1D model across all shells and both metrics, whilst it has higher MSSIM across all shells and lower RMSE within the $b = 2000$ and $b = 3000$ shells compared to SH interpolation. In particular, SH interpolation performance drops relative to all three RCNN models at b-values $b = 2000$ and $b = 3000$. Given that higher b-values yield lower signal-to-noise ratios, this suggests that the deep learning models are potentially more robust to noise, compared to the simpler SH interpolation model. The effect of the shifted distribution at higher b-values can be seen through the difference in RMSE and MSSIM values, independent of model. RMSE, which is not a normalised metric, decreases given an increase in b-value, whereas this relationship is not present within the normalised MSSIM metric.

| | $b = 1000$ | | $b = 2000$ | | $b = 3000$ | |
|---|---|---|---|---|---|---|
| Model | RMSE | MSSIM | RMSE | MSSIM | RMSE | MSSIM |
| SH Interpolation | $65.1 \pm 12.7$ | $0.9854 \pm 0.0044$ | $64.5 \pm 9.1$ | $0.9659 \pm 0.0088$ | $66.7 \pm 13.5$ | $0.9292 \pm 0.0242$ |
| 1D RCNN (Separate) | $70.0 \pm 15.1$ | $0.9852 \pm 0.0054$ | $51.3 \pm 6.7$ | $0.9766 \pm 0.0056$ | $48.1 \pm 8.0$ | $0.9566 \pm 0.0133$ |
| 3D RCNN (Separate) | $\mathbf{63.4 \pm 12.5}$ | $\mathbf{0.9870 \pm 0.0040}$ | $\mathbf{48.1 \pm 6.2}$ | $\mathbf{0.9796 \pm 0.0045}$ | $\mathbf{42.6 \pm 5.5}$ | $\mathbf{0.9633 \pm 0.0088}$ |
| 3D RCNN (Combined) | $65.7 \pm 13.2$ | $0.9869 \pm 0.0041$ | $49.0 \pm 6.0$ | $0.9788 \pm 0.0047$ | $44.8 \pm 6.2$ | $0.9616 \pm 0.0099$ |

Table 2: Average performance of ASR in eight subjects with $q_{in} = 10$ and $q_{out} = 80$. Best results are highlighted in bold.

Figure 2 shows an axial slice of fractional anisotropy (FA) absolute error (AE) within one subject across different models. Here AE is visibly lowest within the 3D RCNN model, whilst the baseline derived AE is lower than in the SH and 1D models. This trend is also true when segmenting the axial slice into WM & grey matter (GM) voxels and analysing the performance separately in each tissue type. The relatively low baseline error is likely due to diffusion tensor imaging (DTI) not requiring HARDI, and therefore is robust even at low q-space sampling rates. Figure B.1 presents the RMSE of the inferred dMRI data used to derive the FA maps, and are consistent with the findings in Figure 2. In particular, the lower relative WM error rates present within the 3D model is important for downstream analysis techniques that require HARDI as they often focus on voxels containing a high proportion of WM. A breakdown of WM and GM RMSE across different subsampling ratios and b-values can be found in appendix B, which contain similar trends as those presented in Tables 1 and 2. WM and GM masks are generated using FSL FAST (Zhang et al., 2001), whilst DTI metrics are generated using the FSL Diffusion Toolbox.

## 4. Conclusion and Future Work

We present a recurrent 3D convolutional architecture to perform angular super-resolution on diffusion MRI data. We compare this methodology against a relevant angular interpolation technique, as well as a 1D variant of the architecture. We demonstrate that the 3D model performs best across various subsampling ratios and b-values. Additionally, we show that this architecture can be used to train a model capable of inferring several different b-

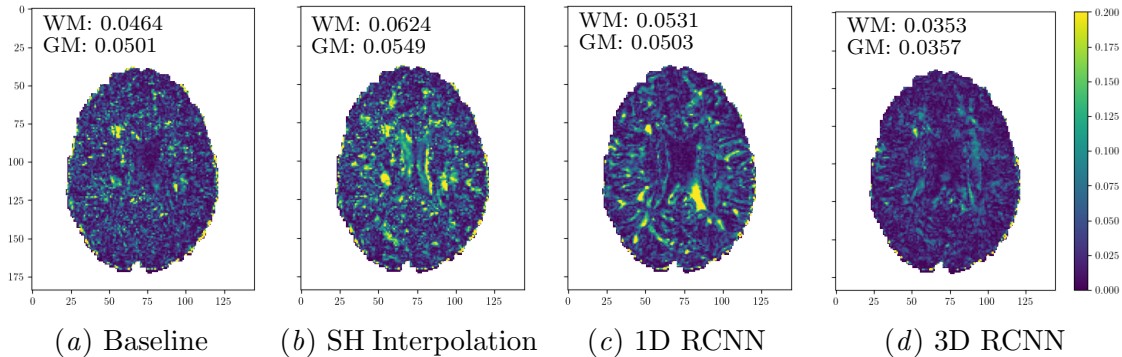

$(a)$ Baseline      $(b)$ SH Interpolation      $(c)$ 1D RCNN      $(d)$ 3D RCNN

Figure 2: Axial slice of FA AE in one subject from the test dataset. ASR is performed with $q_{in} = 6$, $q_{out} = 84$. WM and GM values are averaged across voxels only within the WM and GM mask, respectively. The baseline FA map is calculated from $q_{in}$ volumes whilst other FA maps are derived from both $q_{in}$ and $q_{out}$ data.

values concurrently, albeit at slightly reduced performance compared to individually trained models.

Further work is needed to quantify the robustness of this methodology in out-of-distribution datasets such as those with pathologies and different acquisition parameters, and to provide a comparison with non-recurrent convolutional architectures. A future extension to this work would be expanding the model to explicitly infer other shells, thereby performing multi-shell angular super-resolution. When expanding this model, the effect of the subsampling ratio on multi-shell inference should be explored. Additionally, future work should investigate the effect of angular super-resolution on downstream single-shell and multi-shell analyses that require high angular resolutions.

## Acknowledgments

This work was conducted at the Univeristy of Sheffield whilst authors Matthew Lyon and Mauricio A Álvarez were affiliated with the organisation. This work was funded by the Engineering and Physical Sciences Research Council (EPSRC) Doctoral Training Partnership (DTP) Scholarship. Data were provided [in part] by the HCP, WU-Minn Consortium (Principal Investigators: David Van Essen and Kamil Ugurbil; 1U54MH091657) funded by the 16 NIH Institutes and Centers that support the NIH Blueprint for Neuroscience Research; and by the McDonnell Center for Systems Neuroscience at Washington University.

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

## Appendix A. Model Architecture

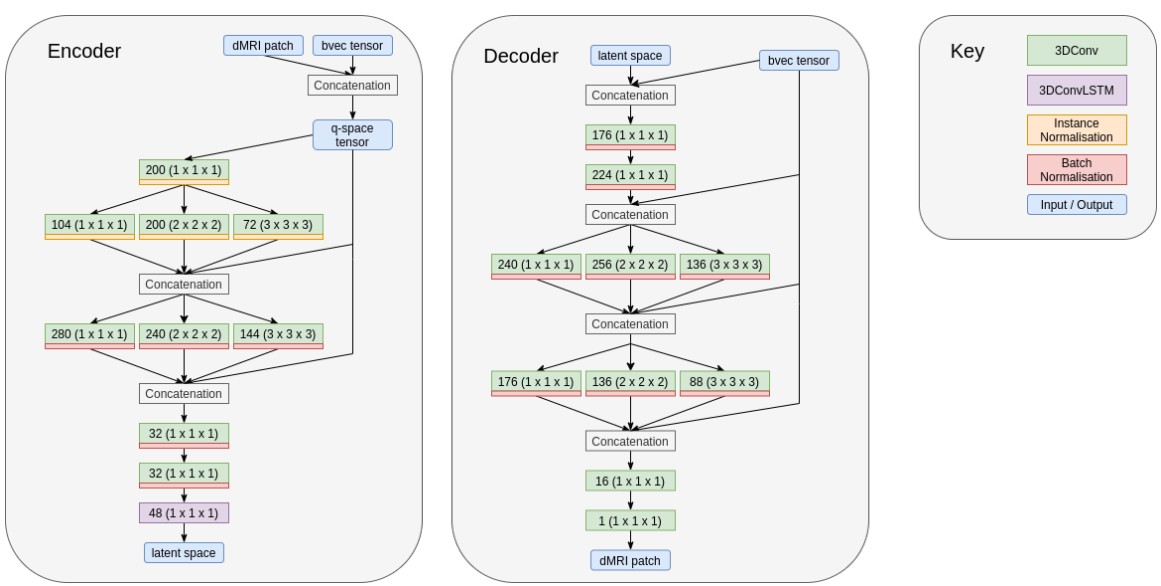

Figure A.1: RCNN model diagram with convolutional filter sizes and channel dimensions. Each convolution node specifies the number of filters used (left) and filter size (right).

## Appendix B. Tissue Specific RMSE

| Model | $q_{in} = 6$, $q_{out} = 84$ | | $q_{in} = 10$, $q_{out} = 80$ | | $q_{in} = 30$, $q_{out} = 60$ | |
|---|---|---|---|---|---|---|
| | WM | GM | WM | GM | WM | GM |
| SH Interpolation | $142.8 \pm 67.9$ | $109.5 \pm 42.8$ | $67.3 \pm 9.7$ | $\mathbf{65.9 \pm 15.2}$ | $63.8 \pm 6.9$ | $\mathbf{65.7 \pm 12.5}$ |
| 1D RCNN | $116.5 \pm 41.5$ | $100.4 \pm 29.2$ | $68.1 \pm 13.2$ | $73.8 \pm 17.1$ | $60.3 \pm 7.9$ | $67.9 \pm 12.3$ |
| 3D RCNN | $\mathbf{82.9 \pm 15.4}$ | $\mathbf{79.1 \pm 14.5}$ | $\mathbf{61.0 \pm 9.1}$ | $66.5 \pm 14.4$ | $\mathbf{59.1 \pm 7.2}$ | $67.3 \pm 11.8$ |

Table B.1: Average WM and GM RMSE in eight subjects with $b = 1000$ across different models. Best results are highlighted in bold.

| Model | $b = 1000$ | | $b = 2000$ | | $b = 3000$ | |
|---|---|---|---|---|---|---|
| | WM | GM | WM | GM | WM | GM |
| SH Interpolation | $67.3 \pm 9.7$ | $\mathbf{65.9 \pm 15.2}$ | $81.1 \pm 12.7$ | $57.1 \pm 8.1$ | $88.0 \pm 19.4$ | $56.1 \pm 10.4$ |
| 1D RCNN (Separate) | $68.1 \pm 13.2$ | $73.8 \pm 17.1$ | $56.2 \pm 6.9$ | $50.6 \pm 7.4$ | $56.7 \pm 10.6$ | $45.6 \pm 7.1$ |
| 3D RCNN (Separate) | $\mathbf{61.0 \pm 9.1}$ | $66.5 \pm 14.4$ | $\mathbf{51.6 \pm 5.7}$ | $\mathbf{48.2 \pm 7.0}$ | $\mathbf{48.5 \pm 7.2}$ | $\mathbf{41.2 \pm 5.1}$ |
| 3D RCNN (Combined) | $62.8 \pm 9.8$ | $69.5 \pm 15.2$ | $52.9 \pm 5.8$ | $48.8 \pm 6.8$ | $52.1 \pm 8.2$ | $42.5 \pm 5.5$ |

Table B.2: Average WM and GM RMSE in eight subjects with $q_{in} = 10$ and $q_{out} = 80$. Best results are highlighted in bold.

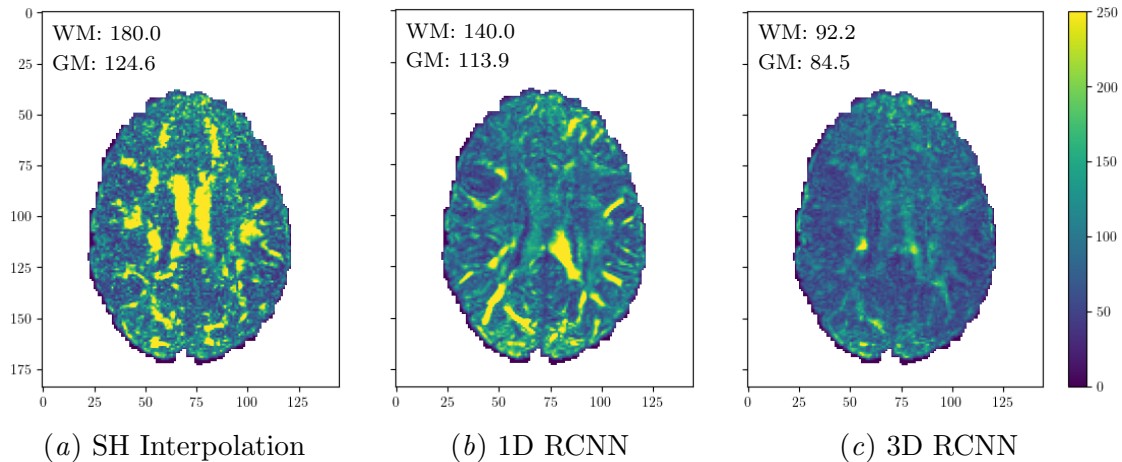

(a) SH Interpolation      (b) 1D RCNN      (c) 3D RCNN

Figure B.1: Axial slice of RMSE in one subject from the test dataset. ASR is performed with $q_{in} = 6$, $q_{out} = 84$. Here each RMSE pixel is the RMSE averaged across $q_{out}$ directions. WM and GM values are the RMSE averaged across spatial voxels and $q_{out}$ directions, for voxels only within the WM and GM mask, respectively.

## Appendix C. Comparison to Non-Recurrent Architecture

We compare performance of the 3D RCNN architecture against a similar 3D CNN design. Here the CNN models use a 1D convolutional layer in place of the ConvLSTM layer, whilst maintaining all other architecture hyperparameters. Table C.1 compares performance across q-space subsampling ratios, whilst Table C.2 compares across b-values. The RCNN performs best across all metrics, suggesting that the added internal complexity of the recurrent layer provides additional capacity for the architecture to capture the non-trivial relationship of q-space in the data.

| Model | $q_{in} = 6, q_{out} = 84$ | | $q_{in} = 10, q_{out} = 80$ | | $q_{in} = 30, q_{out} = 60$ | |
| --- | --- | --- | --- | --- | --- | --- |
| | RMSE | MSSIM | RMSE | MSSIM | RMSE | MSSIM |
| 3D CNN | $84.8 \pm 15.9$ | $0.9758 \pm 0.0085$ | $68.3 \pm 12.6$ | $0.9855 \pm 0.0043$ | $66.4 \pm 10.4$ | $0.9873 \pm 0.0032$ |
| 3D RCNN | $\mathbf{78.4 \pm 13.9}$ | $\mathbf{0.9787 \pm 0.0071}$ | $\mathbf{63.4 \pm 12.5}$ | $\mathbf{0.9870 \pm 0.0040}$ | $\mathbf{63.4 \pm 10.2}$ | $\mathbf{0.9876 \pm 0.0032}$ |

Table C.1: Average performance of ASR in eight subjects with $b = 1000$ across CNN and RCNN models. Best results are highlighted in bold.

| | $b = 1000$ | | $b = 2000$ | | $b = 3000$ | |
|---|---|---|---|---|---|---|
| Model | RMSE | MSSIM | RMSE | MSSIM | RMSE | MSSIM |
| 3D CNN | $68.3 \pm 12.6$ | $0.9855 \pm 0.0043$ | $50.0 \pm 6.3$ | $0.9779 \pm 0.0048$ | $70.7 \pm 12.7$ | $0.9350 \pm 0.0157$ |
| 3D RCNN | $\mathbf{63.4 \pm 12.5}$ | $\mathbf{0.9870 \pm 0.0040}$ | $\mathbf{48.1 \pm 6.2}$ | $\mathbf{0.9796 \pm 0.0045}$ | $\mathbf{42.6 \pm 5.5}$ | $\mathbf{0.9633 \pm 0.0088}$ |

Table C.2: Average performance of ASR in eight subjects with $q_{in} = 10$ and $q_{out} = 80$ in CNN and RCNN models. Best results are highlighted in bold.

