# OpenReview forum: "Angular Super-Resolution in Diffusion MRI with a 3D Recurrent Convolutional Autoencoder"
_MIDL.io/2022/Conference — MIDL 2022_

### Official Review · Reviewer_iTNJ · 2022-01-21

**Confidence:** 5
**Preliminary Rating:** 3
**Recommendation:** Oral, Poster

**Summary:**

HARDI data requires expensive acquisitions due to the high number of diffusion directions. As such, being able to extrapolate higher angular resolution data from lower angular resolution data could reduce acquisition times. The authors propose a 3D RCNN architecture trained on angularly-subsampled HCP data to generate higher angular data. The authors compare themselves to a 1D variant of the same architecture (the dimension being the diffusion signal corresponding to different directions) as well as interpolation from the spherical harmonics representation of the diffusion signal. Two experiments are considered: varying the number of input and output directions and varying the shell used as input and output. Results are convincing and clearly demonstrate the capabilities of the method.

**Strengths:**

As stated previously, the main strength of the paper comes from the results. The performances of the presented method are well presented and superior to the other methods considered. Moreover, the article is very well written: the problem is clearly defined, the motivations for the proposed method are well put forward, the data and its preprocessing are well explained and the architecture of the model and its training procedure are well presented, the discussion surrounding the results is well developed.

**Weaknesses:**

While the paper presented is clearly of high quality, it is in the opinion of the reviewer that the main weakness comes from the thoroughness of the experiments. As it is presented, the contributions are twofold: the use of a recurrent architecture to capture the directional information and the use of a CNN to capture the spatial information. However, only the impacts of CNN are explored in the experiments. It would have been interesting to see if the use of a recurrent network is truly necessary (with the added complexities during training that it brings) to correctly upsample the angular resolution.

Moreover, the analysis of the impacts of the super resolution task is lacking. While Figure 2 shows how much error there is per voxel in a single slice, it would have been interesting to see the effects of this error on DWI-related metrics like (g)FA to assess if the predicted signal is usable in downstream tasks.

**Deanonymize Review:**

no

**Detailed Comments:**

The paper is well written and I have not noticed linguistic or technical errors worth mentioning here.

**Final Rating After The Rebuttal:**

5: Strong Accept

**Justification Of The Final Rating:**

The authors did well in addressing the comments and weaknesses presented in the initial review. Not only did they provide clarifications in the comments and presented work but also provided different results and analyses that, in my opinion, greatly improves the paper and will no doubt be of interest for future readers.

**Paper Type:**

methodological development

**Questions To Address In The Rebuttal:**

The reviewer would like the authors to address the points raised in the "weakness" section. More specifically, mention why was a non-recurrent version of the architecture not mentioned or used as comparison, and if "downstream" analysis of the predicted signal was or could be considered.

**Special Issue:**

no

---

### Official Review · Reviewer_6D6P · 2022-01-23

**Confidence:** 4
**Preliminary Rating:** 3
**Recommendation:** Poster

**Summary:**

The authors proposed a 3D recurrent convolutional autoencoder for super-resolving diffusion MRI in q-space domain. The relationship between q-space samples is modeled using convolutional LSTM cells. It was shown that the proposed approach provided superior performance compared to the conventional SH and 1D variant of the proposed approach.

**Strengths:**

- The paper was presented clearly.
- The use of 3D convolutions for incorporating volumetric spatial relations into modeling
- The addition of LSTM cell to model q-space relationships
- Moving away from SH framework to explore raw dMRI data.


**Weaknesses:**

The comparative study lacks the comparison of the proposed approach with  previous applications. For example, method proposed by Yin et al. (2019) could be added to provide comparison of the proposed approach and its 1D variant.



**Deanonymize Review:**

no

**Detailed Comments:**

It is not clear in the manuscript, how 3D RCNN (Combined) model was implemented and how it is different from the "separate" model. Please elucidate.

**Paper Type:**

methodological development

**Questions To Address In The Rebuttal:**

The method was compared with the conventional method using spherical harmonics. It is necessary to experiment how the proposed approach compares to available learning-based approaches depicted in the manuscript, e.g. Jha et al (2020) and Yin et al (2019).

The effect of network parameters such as the use of parallel convolution blocks needs to be analyzed. For example, how bad the model would perform when only using 3x3x3 convolutions.


**Special Issue:**

no

---

### Official Review · Reviewer_iAVS · 2022-01-23

**Confidence:** 3
**Preliminary Rating:** 5
**Recommendation:** Oral

**Summary:**

Authors present their work on developing an angular super resolution model, trained on the human connectome project diffusion MRI data. They demonstrate that their generator model can create missing direction images, by providing it just 6/90 or 10/90 or 30/90 direction images as input. They compare this to standard spherical harmonics and show better interpolations.

**Strengths:**

This is an interesting topic and application of LSTM, applied to a relevant problem of generating direction images in diffusion imaging. The manuscript is clear and well-written, and of interest to the community.

**Weaknesses:**

The main limitations of the work are:
- it is only applied on healthy adults, which are also relatively young (22 - 35 years, according to PMC3724347). This severely limits the application of this technique in medical imaging, where images are often acquired of patients with brain pathology. Performance of this method in the presence of brain pathology (the most interesting application in my opinion) is therefor unknown.
- only 27 + 3 + 8 = 38 patients out of 1200 (ref: PMC3724347) are used in this study. It is unclear why so few participants from the main study have been included in this work. Furthermore, subject characteristics (age, gender, education, etc) are missing in this work and should be added.
- the effect of the angular super resolution image quality on downstream computations / applications is missing, while I think that would be the most important metrics. Authors provide RMSE and MSSIM values, but to be honest: I find it really difficult to assess what that means in practice. Is this good enough for further applications or still bad?

**Deanonymize Review:**

no

**Detailed Comments:**

How exactly the b-vectors are appended to the patches in the encoder, and provided as a query to the decoder, is not really clear to me. I think a bit more explanation on this would help the reader.

**Final Rating After The Rebuttal:**

5: Strong Accept

**Justification Of The Final Rating:**

No changes were made to the final rating, because I was already satisfied with the answers provided by the authors.
                                                 q

**Paper Type:**

methodological development

**Questions To Address In The Rebuttal:**

- Can authors include more subjects for test?
- Can authors provide subject characteristics and/or how the subject selection from the main study was performed?
- Can authors provide information (either experiments or discussion) on the performance in the presence of brain pathology (ageing lesions such as WMH, vascular lesions, tumours, etc)
- Can authors provide performance metrics on downstream applications? E.g. what is the effect on computing FA / MD ? What is the effect on DTI ? Other applications?

**Special Issue:**

yes

---

### Meta-Review · Area_Chair_5hgt · 2022-02-14

**Recommendation:** Accept (Poster)
**Confidence:** 5

**Metareview:**

This paper proposes a very interesting application of RNNs to super-resolving diffusion MRI data in the angular domain, which is a topic of very high interest in a domain where one is always trying to find a compromise between spatial and angular resolution (number of directions). Despite having some issues here and there, all reviewers were excited about the idea and I am recommending acceptance of the paper.

Having said that: if the authors extend this into a journal article, I would certainly:
1. Compare against stronger methods.
2. Assess the improvements in downstreams tasks, particularly tractography (e.g., can you track fiber crossings that you couldn't resolve without the angular super-resolution?)

---

### Decision · Program_Chairs · 2022-02-28

Accept